# Uncomfortably high: Testing reveals inflated THC potency on retail *Cannabis* labels

**Anna L. Schwabe**[1¤a]*, **Vanessa Johnson**[2], **Joshua Harrelson**[2¤b], **Mitchell E. McGlaughlin**[1]*

**1** University of Northern Colorado, School of Biological Sciences, Greeley, Colorado, United States of America, **2** Mile High Labs, Broomfield, Colorado, United States of America

¤a Current address: Shore Organics/ 420 Organics, Toms River, New Jersey, United States of America
¤b Current address: Cembrex Inc, Longmont, CO, United States of America
* schw0701@bears.unco.edu (ALS); Mitchell.McGalughlin@unco.edu (MEM)

**Data Availability Statement:** All data is in the supplementary table.

**Funding:** Headspace Sensory LLC provided funding for purchase of 13 of the 23 Cannabis

## Abstract

Legal *Cannabis* products in the United States are required to report THC potency (total THC % by dry weight) on packaging, however concerns have been raised that reported THC potency values are inaccurate. Multiple studies have demonstrated that THC potency is a primary factor in determining pricing for *Cannabis* flower, so it has an outsized role in the marketplace. Reports of inflated THC potency and "lab shopping" to obtain higher THC potency results have been circulating for some time, but a side-by-side investigation of the reported potency and flower in the package has not previously been conducted. Using HPLC, we analyzed THC potency in 23 samples from 10 dispensaries throughout the Colorado Front Range and compared the results to the THC potency reported on the packaging. Average observed THC potency was 14.98 +/- 2.23%, which is substantially lower than recent reports summarizing dispensary reported THC potency. The average observed THC potency was 23.1% lower than the lowest label reported values and 35.6% lower than the highest label reported values. Overall, ~70% of the samples were more than 15% lower than the THC potency numbers reported on the label, with three samples having only one half of the reported maximum THC potency. Although the exact source of the discrepancies is difficult to determine, a lack of standardized testing protocols, limited regulatory oversight, and financial incentives to market high THC potency likely play a significant role. Given our results it is urgent that steps are taken to increase label accuracy of *Cannabis* being sold to the public. The lack of accurate reporting of THC potency can have impacts on medical patients controlling dosage, recreational consumers expecting an effect aligned with price, and trust in the industry as a whole. As the legal cannabis market continues to grow, it is essential that the industry moves toward selling products with more accurate labeling.

## Introduction

In the U.S., marijuana is a Schedule I controlled substance and is defined as the flowering tops of *Cannabis sativa* L. containing more than 0.3% total tetrahydrocannabinol (THC) by dry

samples that were included as part of another study [47], but had no other involvement in this study. All other funding was provided by the McGlaughlin Lab at the University of Northern Colorado and by the first author. Mile High Labs provided support for this study in the form of salaries for VJ and JH. The specific roles of these authors are articulated in the 'author contributions' section. The funders had no role in study design, data collection and analysis, decision to publish, or preparation of the manuscript.

**Competing interests:** ALS is an employee of Shore Organics/ 420 Organics; this employment began after the manuscript was completed and submitted for review. VJ and JH were employees of Mile High Labs while data was collected at that facility. JH is employed at Cembrex Inc. There are no patents, products in development or marketed products associated with this research to declare. This does not alter our adherence to PLOS ONE policies on sharing data and materials.

weight [1]. The term "marijuana" has historical negative connotations but there is no alternative agreed upon term for *Cannabis* types with >0.3% THC [2–4], so in the current study all references to *Cannabis* refer to types with >0.3% THC. In recent years attitudes regarding the legal status of *Cannabis* have shifted substantially at the state level with California being the first U.S. state to allow medical *Cannabis* use in 1996, and Colorado and Washington both legalizing retail adult use in 2012. As of July 2022, 37 states in the U.S. have passed legislation allowing medical use, 19 of which also allow legal retail adult use [5]. Nationally, *Cannabis* sales have had significant impacts as evidenced by the almost $10 billion in revenue between 2014 and 2020 in Colorado [6] and record breaking $21.3 billion in national sales in 2020, up 48% from 2019 [7]. Due to the federal status of *Cannabis* as an illegal substance, individual states have had to take on the challenging task of establishing rules for safely producing, testing, selling, and taxing *Cannabis*. As a testament to these challenges, the Colorado Department of Public Health & Environment (CDHPE) does not specify a set of clear testing standards and protocols, but instead suggests using in-house validation for testing protocols using "preferred" methods as they become available, and that novel methods need to be rigorously tested and validated prior to use in the analysis of *Cannabis* product [8].

Female *Cannabis* flowers have densely packed glandular structures called trichomes that store the phytocannabinoids tetrahydrocannabinolic acid (THCA) and cannabidiolic acid (CBDA) which must be decarboxylated by heat to produce Δ9-tetrahydrocannabinol (THC: intoxicating) and cannabidiol (CBD: non-intoxicating). Due to both the intoxicating nature and status as a controlled substance, retail and medical *Cannabis* sold in the legal U.S market requires reporting THC % by dry weight on the product label, either as a range or an average, and may be reported as THC and THCA as separate numbers. The terms "concentration" and "potency" have been used to describe THC % by dry weight; in this work we use potency but recognize that there are many constituents that contribute to the overall effect of a *Cannabis* product. There are thousands of *Cannabis* strains and cultivars with unique chemotypes on the legal market that are often categorized as Sativa, Indica, or Hybrid based on purported effects and available ancestry information. These terms are widely used in the industry, by online databases, and by the community, and although the use of these terms is somewhat disputed [9–11], they will likely continue to be used as part of the common vocabulary for *Cannabis* products.

The medical and retail cannabis industry encourages the production of strains with high THC, as price is often contingent on THC potency [12–15]. Between 1995 and 2012, the THC potency of *Cannabis* seized by the Drug Enforcement Agency (DEA) increased from an average of ~ 4% to ~12% [16], and additional research confirms THC potency has increased consistently since 1970 [17–20]. *Cannabis* available through the legal market is generally reported to have values greater than 15% THC, and as high as 45% in retail market surveys [21]. Studies examining publicly available data [21–23] report higher averages (15%-23.2% THC) than studies conducting independent testing (13.3%-17% THC) [24–27]. Recent online media articles addressing THC potency report strains with THC levels ranging from 21–25% THC, with some strains exceeding 30% [28–31]. Studies relating to THC potency include assessment of how potency influences price [12, 14, 15], how THC content has increased over time [16, 18–20, 32], how cannabinoid profiles vary among samples from different sources [25, 33], and that potency results of the same test material can vary among different testing facilities [34–37].

Many reports have questioned the accuracy of *Cannabis* label data, but limited empirical data exists. A study analyzing sensory perception measured lower than reported THC values from product purchased from two dispensaries in Fort Collins, Colorado (USA) [38], but examining THC potency was not their primary aim. In a recent report, a group of labs in

California collaborated to investigate the extent of the THC potency inflation in 150 randomly chosen flower samples [39]. Although not peer-reviewed, the labs found that 87% of samples had label values outside the permissible >10% deviation and over half with >20% deviation of their reported THC potency [39]. Additionally, an audit of Oregon's testing system in 2019 found that testing may not be reliable and could not ensure safety of the products that were tested, and found some labeled potency results that were substantially higher than the allowable margin of error [40].

The phenomenon referred to as "lab shopping", where cultivators and dispensaries seek out labs that generate the most desirable lab results, is believed to be problematic in many states [39, 41, 42]. Prices for both medical and retail flower are driven by THC concentration, and Cindy Orser of Digipath labs in Nevada places much of the blame for inaccurate reporting on growers and producers who seek out labs that generate higher numbers [42]. Zoorob found that in both Washington and Nevada there is an unusual statistical spike where many tested strains are reported to be just above 20% THC and noted that the spike was most pronounced in data from two testing labs in Washington that recently had their licenses suspended [43]. Cannabis business owners and consumers have also started to recognize that there is potentially inaccurate commercial THC concentration reporting, with lawsuits filed against companies in Arkansas [44] and California [45] alleging that intentional over-representation of THC concentrations in products has occurred to increase profits. Taken together, these recent studies suggest that there is a substantial amount of THC potency inflation in *Cannabis* flower for sale in legal markets.

Colorado Department of Revenue (DOR) licensed testing facilities must follow standard operating procedures outlined in the Cannabis Enforcement Division Colorado Cannabis Rules [46], and labeling affixed to the product for sale must include THC and CBD potency values expressed as a percentage of dry weight. According to the rules, potency should be reported as a range from the lowest to the highest percentage from the Test Batch [46]. Preliminary data collected as part of another study found substantial discrepancies between label reported THC percentages and testing data [25] which prompted a deeper investigation into reported label THC concentrations using third-party test results. In this study, we examined *Cannabis* sourced from across the Colorado Front Range in an attempt to capture a variety of dispensaries with different production protocols and which were presumably using different testing labs to generate the reported THC % dry weight. Sampling was largely opportunistic, including samples collected for other research questions [47] and was intended to provide an overview of label accuracy in Colorado but not to specifically compare dispensaries, cites, or strains. All samples were compared to THC potency results from a single certified third-party *Cannabis* testing lab. This study was designed from a consumer perspective, with all *Cannabis* samples purchased directly from dispensaries and THC values reported on the label were assumed to be accurate. Given the growing economic importance, extensive retail and medical consumption, and proposed medical benefits of *Cannabis*, it is essential that consumers are provided accurate information about the THC potency of *Cannabis* that they purchase for consumption.

## Methods

### Sampling

Sampling was focused on obtaining *Cannabis* flower from a number of Colorado dispensaries to gather general information about the accuracy of THC potency reporting. Samples (n = 23, 1–2 grams per sample) representing 12 strains were purchased from 10 Colorado dispensaries (Table 1). Strains were chosen to represent a diversity of reported proportions of Sativa/Indica

**Table 1. Reported and observed THC % by dry weight from Colorado dispensaries.**

| Strain Name and sample number | Dispensary, License # | City, ZIP code | Reported THC% [1] | Observed Replicates | Observed THC% (Std Dev) | % change[2] | |
|---|---|---|---|---|---|---|---|
| | | | | | | Low | High |
| Colombian Gold '72 | Dispensary 1, 402R-00032 | Denver, 80210 | 19.2 | 6 | 12.98 (1.95) | **-32.40%** | - |
| Blue Dream 1 | Dispensary 1, 402R-00032 | Denver, 80210 | 17.33–33.00 | 2 | 14.91 (0.72) | -14.00% | **-54.80%** |
| Green Crunch (Green Crack) | Dispensary 1, 402R-00032 | Denver, 80210 | 12.80–19.30 | 6 | 14.90 (0.55)[3] | **16.40%** | **-22.80%** |
| Sour Amnesia | Dispensary 1, 402R-00032 | Denver, 80210 | 23.10–26.80 | 6 | 15.19 (1.79) | **-34.20%** | **-43.30%** |
| Mob Boss 1 | Dispensary 1, 402R-00032 | Denver, 80210 | 19.00–31.00 | 3 | 16.37 (0.23) | -13.80% | **-47.20%** |
| Lemon Skunk | Dispensary 1, 402R-00032 | Denver, 80210 | 16.90–17.40 | 6 | 17.65 (2.57) | 4.40% | 1.40% |
| Afghani | Dispensary 2, 402R-00012 | Denver, 80210 | 16.4 | 6 | 11.28 (1.08) | **-31.20%** | - |
| Durban Poison 1 | Dispensary 2, 402R-00012 | Denver, 80210 | 17.4 | 3 | 11.54 (1.38) | **-33.70%** | - |
| OG Kush 2 | Dispensary 2, 402R-00012 | Denver, 80210 | 28.07–31.28 | 2 | 15.71 (0.27) | **-49.80%** | **-56.50%** |
| Bubba 98 | Dispensary 2, 402R-00012 | Denver, 80210 | 24.38 | 5 | 16.40 (3.28) | **-32.70%** | - |
| Durban Poison 4 | Dispensary 3, 403R-00073 | Denver, 80204 | 20.88 | 3 | 13.72 (2.03) | **-34.30%** | - |
| Gorilla Glue #4 | Dispensary 3, 403R-00073 | Denver, 80204 | 25.41–30.88 | 6 | 15.68 (0.77) | **-38.30%** | **-49.20%** |
| Danky Kong | Dispensary 3, 403R-00073 | Denver, 80204 | 26.58–27.73 | 6 | 19.58 (1.92) | **-26.30%** | **-29.40%** |
| Blue Dream 4 | Dispensary 4, 402R-00340 | Denver, 80203 | 14.41–25.18 | 3 | 11.88 (0.53) | **-17.60%** | **-52.80%** |
| Mob Boss 3 | Dispensary 4, 402R-00340 | Denver, 80203 | 22.12–24.87 | 3 | 14.32 (0.74) | **-35.30%** | **-42.40%** |
| Blue Dream 6 | Dispensary 5, 402R-00052 | Garden City, 80631 | 26.65–28.23 | 3 | 15.17 (1.03) | **-43.10%** | **-46.30%** |
| Mob Boss 5 | Dispensary 5, 402R-00052 | Garden City, 80631 | 25.20–28.90 | 3 | 15.54 (0.44) | **-38.20%** | **-46.20%** |
| Durban Poison 5 | Dispensary 6, 402R-00235 | Fort Collins, 80524 | 21.5 | 3 | 13.51 (0.57) | **-37.20%** | - |
| OG Kush 3 | Dispensary 6, 402R-00235 | Fort Collins, 80524 | 24.2 | 3 | 17.86 (0.08) | **-26.20%** | - |
| Blue Dream 5 | Dispensary 7, 402R-00220 | Fort Collins, 80524 | 16.64 | 3 | 10.47 (0.33) | **-37.10%** | - |
| OG Kush 1 | Dispensary 8, 403R-00090 | Denver, 80207 | 15.20–26.14 | 3 | 15.76 (0.35) [3] | 3.70% | **-39.70%** |
| OG Kush 4 | Dispensary 9, 402R-00078 | Denver, 80210 | 15.00–25.01 | 3 | 16.68 (0.28) [3] | 11.20% | **-33.30%** |
| Blue Dream 3 | Dispensary 10, 403R-00596 | Denver, 80239 | 17.87 | 3 | 18.22 (0.32) | 2.00% | - |
| **Mean** | | | **20.27–24.10** | **3.9** | **14.98 (2.23)** | **-23.1%** | **-35.6%**[4] |

[1]—Only some dispensaries reported a range of THC % by dry weight, with both the low and high values shown.

[2]—% Change was calculated as observed THC % dry weight relative to either the lowest or highest reported value. All values that had more than a 15% change are bolded.

[3]—Observed THC % by dry weight fell within the reported range.

[4]—Mean included the low value when only a single value was reported

(100% Sativa– 100% Indica) and reported THC % by dry weight (lowest: 12.8%—highest: 33.0%; Table 1). Sampling was largely opportunistic within three Colorado cities (Denver, Garden City, and Fort Collins) and included samples collected for other studies [25, 47]. All samples were purchased from Colorado licensed dispensaries, which presumably used several different licensed labs for potency testing, although that information was not available at the point of purchase. Label information was recorded as it was printed on the packaging. Some labels reported a range of THC, and some reported a single number. It was unclear if single values reported were an average of multiple tests, or the result of a single test. Procured samples were stored at—4 °C prior to testing. Flower samples were assigned a random number and taken to a third-party lab (Mile High Labs, Loveland CO) for total THC % by dry weight testing using High Performance Liquid Chromatography (HPLC).

## THC testing

Mile High Labs analyzed the samples following standard protocols with a gradient 1260 Infinity II HPLC with DAD detection at 240 nm, (bandwidth 4 nm) and reference 360 nm (bandwidth 100 nm) using a reverse-phase column and guard column with a C18 stationary phase. Cerilliant © Certified Cannabinoid Standards (Redrock, Texas) were used to measure THC and THCA concentrations. Total THC was calculated using the industry standard equation $THC_{TOTAL} = THC + (THCA \times 0.877)$ which determines total THC when THCA and THC are separated during chromatography [46]. All HPLC settings and standards met Colorado state compliance requirements. The dried, homogenized samples were analyzed in three successive HPLC replicates. Additionally, eight samples were analyzed a second time, each with three successive replicates, to test for consistency of the HPLC testing protocol (S1 Table).

## Statistical analyses

Percent change of THC potency was calculated as:

$$\% \ change = \frac{observerd \ potency - reported \ potency}{reported \ potency} * 100\%$$

Percentage change was calculated for both low and high reported THC values when a range was reported on the *Cannabis* packaging. Statistical analyses were conducted using IBM SPSS Statistics for Windows, version 25 (IBM Corp., Armonk, N.Y., USA). An Independent Samples T-test was used to compare the eight samples that were analyzed twice. Due to the differences in the number of samples within our two groups, Reported THC and Observed THC, the nonparametric Wilcoxon Signed Rank Test was used to compare differences between groups. For this analysis only the lowest Reported Label THC value was used.

## Results

Twenty-three *Cannabis* samples representing 12 strains (Table 1) were analyzed. For two samples (Blue Dream 1 and OG Kush 2) one HPLC replicate was excluded from analyses due to poor data. No significant difference was found between the eight samples analyzed in two separate HPLC replicates (Independent Samples T-test, p = 0.240). The Wilcoxon Signed Rank Test, which was conducted using only the first HPLC run for the eight samples that were tested twice, found that there was a significant difference between reported and observed THC % by dry weight (Z = 3.833; p < 0.001; Fig 1). Eighteen of 23 samples (78.26%) had a lower observed THC % by dry weight than the lowest value reported on the label, with an observed average of 14.98% and reported average value of 20.27% and 24.10% for low and high label ranges,

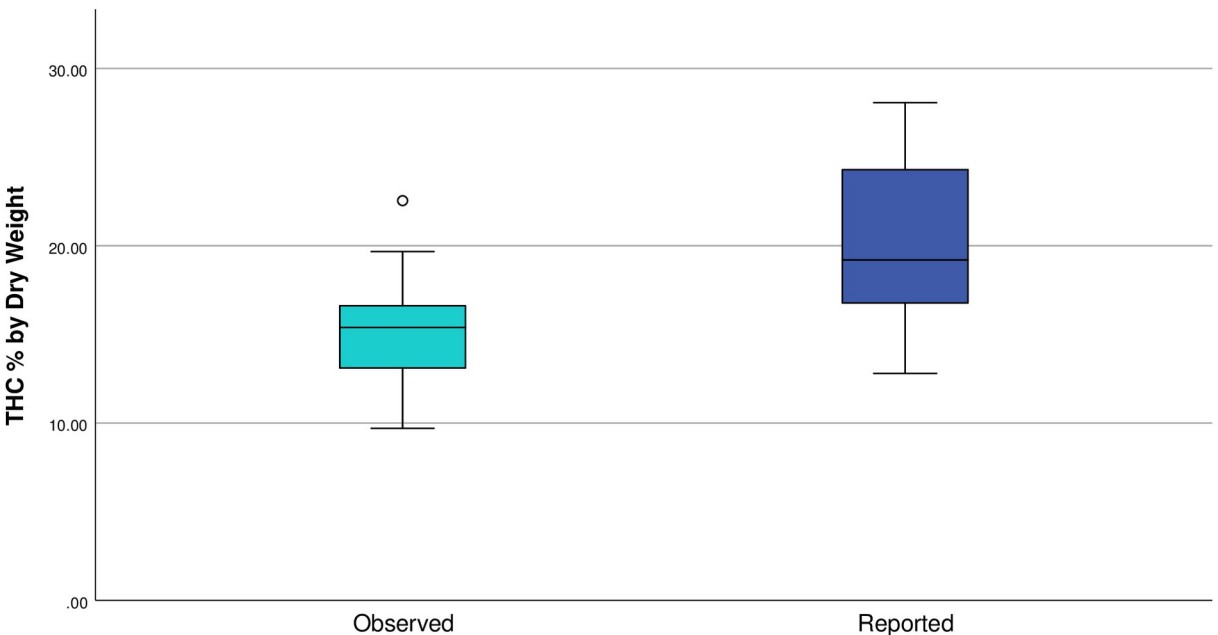

**Fig 1. Box and whisker plot.** Mean THC % by dry weight for observed (light blue) and reported values (dark blue).

respectively (Table 1, Fig 2). Sixteen of 23 samples (69.56%) had observed values that were more than 15% lower relative to the lowest reported THC % by dry weight, and 13 of those samples (56.52%) were more than 30% lower than the reported value. When examining the highest reported value, 20 of 23 samples (86.95%) had observed values that were more than

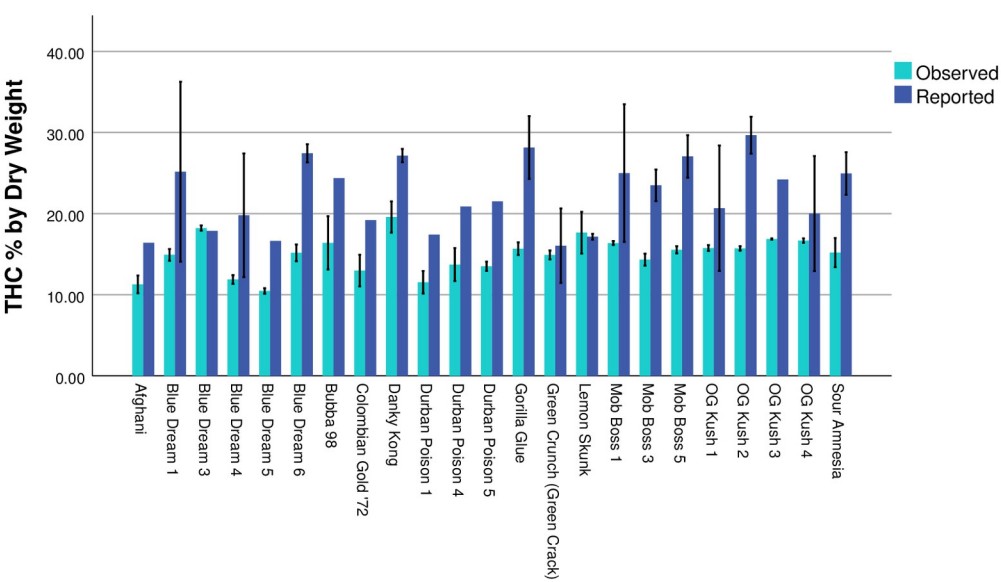

**Fig 2. Mean THC % by dry weight for observed vs. reported values.** Each *Cannabis* sample is indicated along the x-axis, and error bars show the standard deviation. Samples are grouped by the dispensary they were purchased from. For samples with a reported range of THC % by dry weight, the mean of those values is presented. Reported values with no error bar were sold with a single number for THC % by dry weight.

15% lower relative to the highest reported THC %, with 18 of those samples (78.26%) that were more than 30% lower. A single sample, Green Crunch (Green Crack), had an observed value higher than 15% relative to the lowest reported THC % by dry weight, but the label for this sample reported a range that the observed value fell within. Over all samples, the observed THC % by dry weight was 23.1% lower than the lowest reported values and 35.6% lower than the highest reported value.

## Discussion

Our results demonstrate that there are substantial, statistically significant differences between THC % by dry weight (hereafter THC potency) reported on consumer labels and our observed test results (Table 1, Figs 1 and 2). On average, observed THC potency was 23.1% lower than the lowest label reported values and 35.6% lower than the highest label reported values, with a maximum percentage change of -56.5%. Furthermore, 13 of 23 tested samples (~57%) had observed values that were more than 30% lower than the lowest reported value, while only seven samples (~30%) were within 15% of the lowest reported THC potency. Dispensaries with two or more samples had at least one sample where the observed THC potency was more than 34% lower than the lowest label reported value (Table 1). Although we recognize that this study includes a relatively small number of samples, this in no way impacts the empirical test results of those samples, and our sampling included a diversity of dispensaries and strains to provide a board overview of the current state of THC potency reporting along the Colorado Front Range. This is the first peer-reviewed study to empirically examine commercial THC potency; however, there are multiple recent investigative reports from the *Cannabis* industry that support our finding of significant overreporting of THC potency [48, 49], suggesting that overreporting is an industry wide issue. Given that increased THC and CBD potency are associated with higher prices [12–15], and that potency and price are the major factors driving sales [42, 50], comparisons relative to the highest reported potency are likely more indicative of what consumers expect when making purchasing decisions. These results make clear that consumers are often purchasing *Cannabis* that has a much lower THC potency than is advertised and that this occurrence is widespread across the Colorado Front Range.

Information disclosed to the public about THC potency in the legal *Cannabis* marketplace is limited to numbers reported on the label at the point of sale, and some online databases. Dispensaries may also print a QR code on the label which can be scanned to retrieve the Certificate of Analysis (COA) lab report detailing cannabinoid and other tests performed on the batch the sample came from. However, raw data and testing methods are not often included on packaging or in COA reports. Recent published studies report retail THC potency ranging from ~15–23.2% [21–25], while online articles reference strains with THC potency ranging from 21–25%, with some strains with >30% THC potency [28–31]. Vergara et al. [23] found differences in average THC levels among states, which may indicate different testing methods, but may also reflect regional consumer preferences. An investigative study conducted in 2011 tested six samples at ten different labs and found the testing results deviated substantially for each of the samples [51]. ElSohly et al. [18] evaluated 14,234 samples of *Cannabis* from 2009 to 2019 and the results showed the mean THC potency had increased from 9.75% in 2009 to 13.88% in 2019, which is in line with our observed average of 14.98%, as opposed to the average THC potency advertised online for medicinal (19.2% ±6.2) and retail (21.5% ±6.0) products [21].

Although our study was not designed to pinpoint the source of THC potency discrepancies, we have outlined four areas that are likely to be important based on Colorado regulations.

1. Sample Collection–It has been documented that THC potency varies in flowers located on the top, middle and lower branches on the plant as well as the timing of plant development

[52]. Colorado has developed guidelines for how flowers should be sampled [53], but there is limited to no enforcement of these guidelines. If growers are not randomly selecting flowers from throughout plants for testing, THC potency results may not be indicative of the entire batch [52].

2. Sample Testing—Although testing facilities are licensed through the DOR and standard operating procedures have been outlined by the Cannabis Enforcement Division Colorado Cannabis Rules [45], there are multiple variables that could lead to different results. Of note, Colorado standard operating procedures allow for differences in sample preparation, usage of different internal standards, and usage of different analytical instruments, all of which could impact THC potency measurements. The testing instrumentation used in this study is the most common method used in Colorado, with all ten current testing labs advertising HPLC as the method to quantify the THC % by dry weight of cannabinoids tested in each sample. Therefore, the methodology used in this study should not impact the observed decrease in THC content of samples.

3. Sample Degradation–THC is known to degrade if not stored correctly. Ross and ElSohly [54] found that when stored at room temperature THC potency decreased by 16.6% (±7.4) after one year, and up to 41.4% (±6.5) after four years. Furthermore, when exposed to light at room temperature, THC is almost 100% degraded after four years [55]. However, when THC degrades, it is converted to cannabinol (CBN) which was not observed in sizeable quantities in the samples used in this study, indicating the lower potency in the observed versus reported values were not due to age or poor storage conditions.

4. Economic Incentive–Consumers often make purchasing decisions based on THC potency and are willing to pay more for higher potency flower [12, 50, 56]. This creates incentive for retailers and testing facilities to report the highest possible THC potency, which could be inflated by manipulating flower sampling, analytical procedures, or reported data. The research by Zoorob (2021) clearly illustrates that reported THC potency test results in Nevada and Washington show an unusually high frequency of products with potency just above 20% [43]. Additionally, media outlets have released reports of testing labs in multiple states (California, Colorado, Oregon, Washington, and Nevada) that had misrepresented potency values [57–59].

It is currently unclear how many consumers are aware of issues with THC potency reporting and how it might impact their purchasing decisions. Our work only examined flower material but given the financial incentives to report high THC potency it is likely that inflated potency is common for all THC containing products. Accurate label reporting is essential to ensure consumer trust. If consumers find they cannot trust THC potency reporting, they may also question the reliability of other testing results, such as a lack of pesticides, molds, and other contaminants [60].

Based on our results, there are several areas where THC potency testing and reporting could be improved. In Colorado, *Cannabis* flower sold on the legal market is required to be tested for total THC potency by a licensed testing facility that is reported as a range on the label [45]. Only 61.9% of our samples had a reported range of THC potency and in most samples that range was much higher than our observed values. This suggests that both flower harvesting protocols and reporting rules need to be enhanced to ensure commercial samples fall within the reported range. Independent testing should be implemented to compare THC potency values at time of harvest and point of sale to determine if THC potency is changing throughout the processing and sale. Finally, rigorous standard operating procedures should be developed to minimize testing variation among labs.

## Conclusion

As *Cannabis* becomes more widely used in retail and medicinal contexts, it is important that consumers are presented with accurate information about the THC potency of what they are consuming. Our results clearly demonstrate that retail *Cannabis* flower THC potency is significantly inflated in samples purchased in Colorado. Given the numerous recent reports and lawsuits questioning THC potency reporting, it is likely that this is an industry wide problem. Additional studies should be conducted in other states and with larger sample sizes to confirm our findings. Although we have no power to change the current system, we hope highlighting this issue and educating consumers will affect the change needed to remedy inflated potency of flower products. Addressing this discrepancy will require both changes to the regulatory system and consumer awareness that reported THC potencies are frequently inflated.

## Supporting information

**S1 Table. Raw data from HPLC testing conducted by Mile High Labs.**
(PDF)

## Acknowledgments

We would like to thank Laura Heiker for consulting on regulatory and compliance for Colorado testing procedures, Emily Holt and Scott Franklin for aiding with the statistical methods, and Gabriella Mueller and Cheysser Harding for arranging the HPLC testing.

## Author Contributions

**Conceptualization:** Anna L. Schwabe, Mitchell E. McGlaughlin.

**Funding acquisition:** Anna L. Schwabe, Mitchell E. McGlaughlin.

**Methodology:** Anna L. Schwabe, Vanessa Johnson, Joshua Harrelson.

**Supervision:** Anna L. Schwabe, Mitchell E. McGlaughlin.

**Writing – original draft:** Anna L. Schwabe, Mitchell E. McGlaughlin.

**Writing – review & editing:** Anna L. Schwabe, Vanessa Johnson, Mitchell E. McGlaughlin.

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
