## [Decision Letter · Decision Letter 0]

14 Jul 2022

PONE-D-22-13833

Uncomfortably High: Testing Reveals Inflated THC Potency on Retail Cannabis Labels

PLOS ONE

Dear Dr. McGlaughlin,

Thank you for submitting your manuscript to PLOS ONE. After careful consideration, we feel that it has merit but does not fully meet PLOS ONE’s publication criteria as it currently stands. Therefore, we invite you to submit a revised version of the manuscript that addresses the points raised during the review process.

The abstract should be more detailed and the introduction should include appropriate references of similar studies that have already been published.De-identification of the dispensaries from which product for testing was obtained unless the authors are able to establish consent from the stated dispensaries.Expended details on testing methods both by the authors and the methods used by the dispensaries, highlighting any differences.Sample size justification

It is also suggested that the term recreational cannabis be changed to non-medical use of cannabis.

We look forward to receiving your revised manuscript.

Kind regards,

Lori A. Walker

Academic Editor

PLOS ONE

Journal Requirements:

   "Acknowledgments: We would like to thank Laura Heiker for consulting us on regulatory and compliance for Colorado testing procedures, Emily Holt and Scott Franklin for aiding with the statistical methods, Gabriella Mueller and Cheysser Harding for arranging the HPLC testing, and Avery Gilbert for providing some of the funding to purchase samples

Funding: Funding was provided by the University of Northern Colorado Graduate Student Association, the McGlaughlin Lab at the University of Northern Colorado, the first author and Headspace Sensory LLC."

 "The authors received no specific funding for this work"

Reviewers' comments:

Reviewer's Responses to Questions

**Comments to the Author**

1. Is the manuscript technically sound, and do the data support the conclusions?

Reviewer #1: Yes

Reviewer #2: Partly

2. Has the statistical analysis been performed appropriately and rigorously? 

Reviewer #1: Yes

Reviewer #2: No

3. Have the authors made all data underlying the findings in their manuscript fully available?

Reviewer #1: Yes

Reviewer #2: Yes

4. Is the manuscript presented in an intelligible fashion and written in standard English?

Reviewer #1: Yes

Reviewer #2: Yes

5. Review Comments to the Author

Reviewer #1: This is a thoughtful paper that is well written. My only comment is that the implications for public health policy should be drawn out more fully in the discussion. What does this mean for keeping consumers in a potentially "safer" regulated and legal market? While accurate THC potency is a quality control issue, is is also a consumer trust issue and I think there would be concerns (beyond profit motives) about losing people to an illicit market? Finally, while this paper addresses flower product I think it requires some thought about implications for shatter and dabs - the products typically associated with the highest THC potency. Overall, great paper! I think it will have an impact.

Reviewer #2: The issue described in this manuscript, specifically the listed potency of delta-9 THC products commercially and their actual potency, is an important question and standardized testing or regulations around testing have not been formalized. While this is an important topic to study, the gaps in the literature are not well articulated given that other studies have analyzed commercially available products and have demonstrated discrepancies in listed vs. actual values.

There has been an emphasis on referring to recreational cannabis use as non-medical use (as compared to medical use). The authors may consider use of these terms in their manuscript.

Is including the names of dispensaries appropriate? It seems that the information should be coded to maintain confidentiality, even though these were purchased legally by investigators. Given that samples were purchased there without dispensaries consenting to research, can the authors state the ethical conduct of this study and approval by the institution?

How does the testing of product completed as part of this study compare to how dispensaries may be testing product? The specifics of testing among sampled dispensaries is not listed, but some general sense of how these tests are normally being conducted would be helpful.

The justification for 23 samples is not provided and no justification is provided for the number selected from each city and from each dispensary.

Since samples varied across cities and dispensaries, were there relationships between accuracy across those variables? Would it be the case that certain dispensaries, given their testing standards, would be more accurate than others, or does strain play a role in this?

Minor comments:

The abstract is very short and additional information could be included. I believe PLOS One has an abstract word count of 300 words, so details could be expanded on this study. In expanding the information in the abstract, abbreviations should also be defined upon first use.

6. PLOS authors have the option to publish the peer review history of their article (what does this mean?). If published, this will include your full peer review and any attached files.

Reviewer #1: No

Reviewer #2: No

---

## [Author Response · Author response to Decision Letter 0]

17 Aug 2022

Please see the response to reviewers with information related to editor and reviewer comments.

Academic Editor Comments

• The abstract should be more detailed and the introduction should include appropriate references of similar studies that have already been published.

- We have substantially expanded the abstract to add more background and context to our findings. We have also added more background to introduction, particularly related to previous THC potency studies (particularly LN 88-114), and expanded our literature cited throughout the manuscript.

• De-identification of the dispensaries from which product for testing was obtained unless the authors are able to establish consent from the stated dispensaries.

- All dispensary information has been de-identified.

• Expended details on testing methods both by the authors and the methods used by the dispensaries, highlighting any differences.

- We have provided additional context on how our HPLC methods relate to the testing labs our conducting. As we have now clarified, we used the method that is advertised by all current testing facilities in the state of Colorado (LN 261-265) and our analytical methods were developed based on State of Colorado testing guidelines (LN 159-160).

• Sample size justification

- We have provided more context on our sampling detailing that it was largely opportunistic (LN 126-128, 141-142). This study came about when we were collecting data for a sensory project and we discovered that there was a large discrepancy between reported and observed potency. Based on this unexpected result we validated our methods by retesting 8 samples, as detailed in the methods, and expanded our sampling as resources allowed. Although our sample size is relatively small, that in no way impacts the result that there is substantial over-reporting of potency occurring in the marketplace (LN 222-225).

It is also suggested that the term recreational cannabis be changed to non-medical use of cannabis.

- We do not like the term ‘non-medical use’, but we have replaced all references to ‘recreational’ to ‘retail” as referred to by the Colorado Department of Revenue Cannabis Enforcement Division.

Reviewer 1 Comments

Reviewer #1: This is a thoughtful paper that is well written. My only comment is that the implications for public health policy should be drawn out more fully in the discussion. What does this mean for keeping consumers in a potentially "safer" regulated and legal market? While accurate THC potency is a quality control issue, is is also a consumer trust issue and I think there would be concerns (beyond profit motives) about losing people to an illicit market?

- We have added more context related to public health in the discussion specifically addressing public trust (LN 282-288).

Finally, while this paper addresses flower product I think it requires some thought about implications for shatter and dabs - the products typically associated with the highest THC potency.

- Although the question of accuracy related to reported potency for extracts and concentrates is of interest, it is outside the scope of the current work. We did suggest that additional THC containing products should be tested in a similar study (LN 283-285) and included reference to one review paper that summarizes what is known about label accuracy for other products (LN 288).

Reviewer 2 Comments

Reviewer #2: The issue described in this manuscript, specifically the listed potency of delta-9 THC products commercially and their actual potency, is an important question and standardized testing or regulations around testing have not been formalized. While this is an important topic to study, the gaps in the literature are not well articulated given that other studies have analyzed commercially available products and have demonstrated discrepancies in listed vs. actual values

- We have expanded our introduction to cover more recent publications related to listed vs. actual potency values (LN 88-98) and reports of “lab shopping” (LN 100-114). However, this continues to be an understudied area with very limited studies that have collected empirical data.

It seems that the information should be coded to maintain confidentiality, even though these were purchased legally by investigators. Given that samples were purchased there without dispensaries consenting to research, can the authors state the ethical conduct of this study and approval by the institution?

- All dispensaries have been de-identified.

How does the testing of product completed as part of this study compare to how dispensaries may be testing product? The specifics of testing among sampled dispensaries is not listed, but some general sense of how these tests are normally being conducted would be helpful.

- We have expanded our description of how testing is normally done in Colorado and that HPLC is the most common method (LN 261-265). 

The justification for 23 samples is not provided and no justification is provided for the number selected from each city and from each dispensary.

- We have provided more context on our sampling detailing that it was largely opportunistic (LN 126-128, 141-142). This study came about when we were collecting data for a sensory project and we discovered that there was a large discrepancy between reported and observed potency. Based on this unexpected result we validated our methods by retesting 8 samples, as detailed in the methods, and expanded our sampling as resources allowed. Although our sample size is relatively small, that in no way impacts the result that there is substantial over-reporting of potency occurring in the marketplace (LN 222-225). We have also reorganized Table 1 and Figure 2 so samples are grouped by dispensary and discussed that the observed discrepancies occurred for all dispensaries with 2 or more samples in the dataset (LN 220-222).

---

## [Decision Letter · Decision Letter 1]

3 Nov 2022

PONE-D-22-13833R1

Uncomfortably High: Testing Reveals Inflated THC Potency on Retail Cannabis Labels

PLOS ONE

Dear Dr. McGlaughlin,

Thank you for submitting your manuscript to PLOS ONE. After careful consideration, we feel that it has merit but does not fully meet PLOS ONE’s publication criteria as it currently stands. Therefore, we invite you to submit a revised version of the manuscript that addresses the points raised as noted below.

We look forward to receiving your revised manuscript.

Kind regards,

Lori A. Walker

Academic Editor

PLOS ONE

***Request from Staff Editors***

In the interest of reproducibility and to ensure that your submission meets our third publication criterion https://journals.plos.org/plosone/s/criteria-for-publication#loc-3 please address the following points noted below:

We understand that you purchased cannabis from local dispensaries for this study. In your Methods section, please provide additional details regarding the source of this material assayed. For instance, if available, please provide the geographic coordinates and/or names of the purchase locations/dispensaries (e.g., stores, markets), as well as any further details about the purchased items (e.g., lot number, source origin, description of appearance).  

In addition, please provide further details about the criteria used to select the dispensaries from which to assay samples.  

Lastly please consider expanding on the limitations of the study, including on the generalisability of the findings for all retail Cannabis labels as inferred from the title of this work.  

Journal Requirements:

Reviewers' comments:

Reviewer's Responses to Questions

**Comments to the Author**

1. If the authors have adequately addressed your comments raised in a previous round of review and you feel that this manuscript is now acceptable for publication, you may indicate that here to bypass the “Comments to the Author” section, enter your conflict of interest statement in the “Confidential to Editor” section, and submit your "Accept" recommendation.

Reviewer #2: All comments have been addressed

Reviewer #3: All comments have been addressed

2. Is the manuscript technically sound, and do the data support the conclusions?

Reviewer #2: Yes

Reviewer #3: Yes

3. Has the statistical analysis been performed appropriately and rigorously? 

Reviewer #2: Yes

Reviewer #3: Yes

4. Have the authors made all data underlying the findings in their manuscript fully available?

Reviewer #2: No

Reviewer #3: Yes

5. Is the manuscript presented in an intelligible fashion and written in standard English?

Reviewer #2: Yes

Reviewer #3: Yes

6. Review Comments to the Author

Reviewer #2: Revisions are satisfactory and I recommend acceptance. Data availability is unclear though in the statement provided by the authors and should be clarified per journal requirements.

Reviewer #3: The manuscript is well written, the data are of public and scientific interest, the results interpretation is balanced and on point.

Consider adding some comments that THC concentrations could also have a different meaning depending on the potential CBD presence, concentration, and ratio.

7. PLOS authors have the option to publish the peer review history of their article (what does this mean?). If published, this will include your full peer review and any attached files.

Reviewer #2: No

Reviewer #3: No

---

## [Author Response · Author response to Decision Letter 1]

6 Dec 2022

December 6, 2022

Dear Dr. Walker,

Thank you for the recent reviews on our manuscript PONE-D-22-13833R1, Uncomfortably High: Testing Reveals Inflated THC Potency on Retail Cannabis Labels. We have responded to all of the staff editor and reviewer comments, as detailed below, and submitted an updated version of the manuscript with and without track changes. Overall, we feel that the manuscript is ready for publication. If you have any questions or concerns, feel free to contact us.

Sincerely,

Mitchell McGlaughlin and Anna Schwabe

• We understand that you purchased cannabis from local dispensaries for this study. In your Methods section, please provide additional details regarding the source of this material assayed. For instance, if available, please provide the geographic coordinates and/or names of the purchase locations/dispensaries (e.g., stores, markets), as well as any further details about the purchased items (e.g., lot number, source origin, description of appearance).

To address this comment, we did two things. 

1) We added dispensary license numbers and zip codes to Table 1 (Ln 225). As we previously communicated to the editorial staff (but never received a response), we do not want to list dispensaries by name because our goal was not single out ‘good’ or ‘bad’ dispensaries, rather we wanted to show that there are THC potency reporting issues across many producers. Removing dispensary names was also requested by a reviewer during the review process. We feel that the details in Table 1 now strike the appropriate balance between our goals and documenting where sampling occurred.

2) We reworked our sampling details in the methods (Ln-151-158). Since we were looking at broad patterns there was not a specific sampling strategy beyond having samples with a diversity of Sativa/Indica proportions and reported THC concentrations (ln 154-155).

We do not feel that descriptions of the material, beyond that we purchased flower, is necessary. This is particularly relevant since we approached this study via a consumer perspective. Other sample details, including lot # or source, are incredibly inconsistent from producers, making them not informative. These reporting issues are highlighted throughout our manuscript and in multiple cited studies within.

• In addition, please provide further details about the criteria used to select the dispensaries from which to assay samples.

Addressed relative to the previous requested edit. Our sampling strategy is addressed in the introduction (Ln 121-126) and methods (Ln 136-137, 141-145). 

• Lastly please consider expanding on the limitations of the study, including on the generalisability of the findings for all retail Cannabis labels as inferred from the title of this work.

We previously had content that addressed our sample size limitations (Ln 244-247). In this version we have expanded upon that discussion (Ln 248-251) providing additional citations that support that our findings are consistent with what is occurring across the industry. We also added additional text to the introduction (Ln 108-112) to include a second lawsuit related to inflated THC potency and then adjusted our conclusions section (Ln 340-343) to make clear that our results are specific to samples in Colorado and to recommend further research including more samples from other states.

Reviewer Comments

Reviewer #2: Revisions are satisfactory and I recommend acceptance. Data availability is unclear though in the statement provided by the authors and should be clarified per journal requirements.

All of our data is available in supplementary Table S1 as stated on Ln 575 and uploaded into the manuscript system.

Reviewer #3: The manuscript is well written, the data are of public and scientific interest, the results interpretation is balanced and on point.

Consider adding some comments that THC concentrations could also have a different meaning depending on the potential CBD presence, concentration, and ratio.

This request is outside the scope of our manuscript. The reviewer is absolutely correct that there are different effects and descriptions based on combinations of different cannabinoids. This is an active area of study within the field, which is best approached with different methods and study design. All samples included in this study were high THC drug-type samples available at recreational Cannabis dispensaries. We make clear that we are looking solely at THC % throughout the introduction based on the importance of THC % to regulation/defining drug-type Cannabis (Ln 43-44) and the importance it plays in marketing and pricing Cannabis (Ln 74-112).

---

## [Editor Report · Decision Letter 2]

15 Feb 2023

Uncomfortably High: Testing Reveals Inflated THC Potency on Retail Cannabis Labels

PONE-D-22-13833R2

Dear Dr. McGlaughlin,

We’re pleased to inform you that your manuscript has been judged scientifically suitable for publication and will be formally accepted for publication once it meets all outstanding technical requirements.

Kind regards,

Lori A. Walker

Academic Editor

PLOS ONE
---

## [Editor Report · Acceptance letter]

16 Mar 2023

PONE-D-22-13833R2 

Uncomfortably High: Testing Reveals Inflated THC Potency on Retail *Cannabis* Labels 

Dear Dr. McGlaughlin:

I'm pleased to inform you that your manuscript has been deemed suitable for publication in PLOS ONE. Congratulations! Your manuscript is now with our production department. 

Kind regards, 

on behalf of

Dr. Lori A. Walker 

Academic Editor

PLOS ONE